# Synthesis of Carbon Microspheres from Inedible Crystallized Date Palm Molasses: Influence of Temperature and Reaction Time

**DOI:** 10.3390/ma16041672

**Published:** 2023-02-16

**Authors:** Mohanad El-Harbawi, Saeed Alhawtali, Abdulrhman S. Al-Awadi, Lahssen El Blidi, Maher M. Alrashed, Abdulrahman Alzobidi, Chun-Yang Yin

**Affiliations:** 1Department of Chemical Engineering, King Saud University, Riyadh 11421, Saudi Arabia; 2K.A. CARE Energy Research and Innovation Center in Riyadh, King Saud University, Riyadh 11421, Saudi Arabia; 3Newcastle University in Singapore, 537 Clementi Road #06-01, SIT Building @ Ngee Ann Polytechnic, Singapore 599493, Singapore

**Keywords:** inedible crystallized date palm molasses, hydrothermal carbonization, carbon microspheres, temperature, reaction time, yield

## Abstract

In this work, carbon microspheres (CMs) were prepared by hydrothermal carbonization (HTC) of inedible crystallized date palm molasses. The effects of temperature and reaction time on the prepared materials were studied. Experiments were carried out at different temperatures (180, 200, 230 and 250 °C) with reaction times ranging from 2 to 10 h. It was found that temperature had the greatest influence on the mass yield of the CMs. No solid products were observed at a temperature of 180 °C and a reaction time less than 2 h. The highest yield was found to be 40.4% at 250 °C and a reaction time of 6 h. The results show that the CMs produced were approximately 5–9 μm in diameter. The results also show that the largest diameter of the CMs (8.9 μm) was obtained at a temperature of 250 °C and a reaction time of 6 h. Nonetheless, if the reaction time was extended beyond 6 h at 250 °C, the CMs fused and their shapes were deformed (non-spherical shapes). The synthesized materials were characterized using Scanning Electron Microscopy (SEM), Fourier Transform Infrared (FTIR), Branuer-Emmett-Teller (BET) and thermogravimetric analysis (TGA). BET surface areas for the four samples were found to be less than 1 m^2^/g. The methylene blue adsorption studies indicated that the equilibrium adsorption capacity was reached after 15 min, with a maximum adsorption capacity of 12 mg/g. The recycling of date palm molasses (a known processed waste) to generate a useable carbon microsphere represents a beneficial step in the application of sustainable processing industries in the Middle East.

## 1. Introduction

Micro-sized carbonaceous materials have promising applications in many fields, including carbon fixation, water purification, fuel cell catalysis, energy storage, bioimaging, drug delivery and gas sensors [1,2]. In particular, there is significant interest in the production of colloidal carbon microspheres (CMs) of numerous sizes by hydrothermal carbonization (HTC). Such products are similar to coal produced from the geological process of “coalification” beneath the Earth’s surface, which happens through the conversion of natural carbon-containing compounds under high pressure and temperature [3].

HTC is a process in which biomass is treated in a closed aqueous system at temperatures ranging from 180 to 250 °C and self-generated pressure to generate solid materials known as hydrochar [4]. In general, hydrothermal CMs can be easily produced at relatively low temperatures and pressures by digesting low-cost available organic precursors, such as saccharides (glucose, sucrose and fructose) or biomass. This approach is simple, cost-effective and environmentally friendly. As well as utilizing biomass waste to produce useful microparticles, this approach affords a distinctive carbon sequestration technology by eliminating carbon derived from plant material from the short-term carbon cycle.

HTC of various types of saccharides has been studied by several researchers. Cai and co-researchers [5] synthesized CMs from glucose at 130 °C for 12 h in the presence of aluminum chloride. Qi and co-researchers [6] successfully obtained CMs from glucose at 180 °C for 10 h, while Demir-Cakan and co-researchers [7] synthesized CMs from glucose at 190 °C for 16 h in the presence of acrylic acid. In another study, Li and co-researchers [8] investigated the effects of various parameters on the preparation of CMs from glucose, including glucose concentration (0.3–0.5 mol/L, 50 mL), reaction temperature (140–210 °C) and residence time (3–7 h). The results showed that temperatures between 180 and 190 °C and a reaction time of 4–5 h were suitable for the formation of CMs. Interestingly, they found that, at a reaction time of 7 h, the produced material melted and formed peanut-like and other irregular shapes. Sanchez-Sanchez et al. [9] synthesized CMs based on tannin–sucrose mixtures using different pH (2 to 8) and temperatures (160, 180 or 200 °C). The results showed that the yield of CMs increased with increasing pH, while a temperature increase from 160 to 200 °C slightly decreased the yield of CMs. Furthermore, the results revealed that the addition of more sucrose resulted in smaller microspheres, which was due to the increased amount of spores produced by the HTC of sucrose. Sulistya et al. [10] prepared CMs from sucrose with citric acid at 190 °C and for 6 h. Bedin et al. [11] concluded that the optimal conditions for obtaining CMs from sucrose are at a temperature of 194 °C, a reaction time of 1197 min (≈50 h) and a sucrose concentration of 0.85 mol/L. Zhao and co-researchers [12] and Shi and co-researchers [13] obtained CMs from HTC of sucrose at 180 °C with reaction times of 12 h and 24 h, respectively. In another study, Zhang and co-researchers [14] used fructose as a starting material for the synthesis of CMs using HTC at temperatures ranging from 150 to 190 °C for various reaction times (up to 48 h). Findings revealed that the average size and the quantity of the CMs increased with increasing temperature and reaction time. Jung and co-researchers [15] studied the growth and formation rates of CMs prepared from fructose at 200 °C and between 20–180 min reaction time using HTC with three additive salts; namely, KCl, CaCl_2_ or FeCl_3_. The results indicated that the particles achieved stable particle size at longer reaction times, regardless of the added salts. Ryu and co-researchers [16] found that the carbon productivity from these sugars was very low; for instance, the yields of the solid products derived from xylose and fructose were only 3.2%. It was reported that the temperature for the onset of HTC of sugars, such as glucose, sucrose, and fructose (~160–170 °C), was lower than that for cellulose [4,17]. This is attributed to a high degree of structural complexity, which is thought to affect the mechanism of HTC [18,19]. However, in another study, solid carbon spheres were not obtained at temperatures lower than 180 °C [8]. Li and co-researchers [8] found that temperatures below 180 °C were most likely insufficient to break down the major components of sugar, and no solid products were formed. In contrast, a solid product is formed at temperatures higher than 180 °C [8]. This may indicate that temperatures between 180 and 250 °C are suitable for the formation of the solid products. Nevertheless, it is worth noting that the use of high temperatures with sufficient reaction time may lead to the formation of secondary char, which dominates the mechanisms of hydrochar formation [20].

Date palm (*Phoenix dactylifera* L.) is a prominent harvest plant in Saudi Arabia as well as the greater Middle East region. There are more than 31 million palm date trees in the Kingdom of Saudi Arabia, making the country the second largest producer of dates in the world, producing about 1.07 million tons of dates per year [21]. Large quantities of these produced dates are wasted due to long or improper storage. Therefore, many date producers resort to turning them into molasses. Date molasses or date syrup has a shelf life of about two years if stored under suitable conditions. After this period, it begins to crystallize and becomes inedible and must be discarded. Date molasses consists of large amounts of reducing sugars. Glucose and fructose are the two main components of sugar in date molasses [22]. “Sukkari” type palm trees account for the largest proportion in the Kingdom of Saudi Arabia. The fruits of these trees contain glucose (51.80%), fructose (47.50%) and a small amount of sucrose, fructose and galacturonic acid [23]. To the best of our knowledge, the production of CMs from date molasses by hydrothermal carbonization method has not been previously reported. As such, our main motivation in the current study was to use the crystallized and inedible palm molasses (waste molasses) and convert them to CMs, as they possessed high contents of glucose and fructose. In the study, we also investigated the effect of temperature and reaction time on the formation and production rate of CMs. Finally, we examined the ability of the as-synthesized CMs to remove methylene blue, a known water-polluting dye.

## 2. Materials and Methods

### 2.1. Raw Material and Chemicals

Crystallized, inedible, expired molasses was obtained from a local market in Riyadh. Deionized water was used for the HTC process and for washing the solid samples. Methylene blue with a purity of 95% was purchased from LOBA Chemie, India. Absolute ethanol (99.9%) was purchased from Sigma-Aldrich, U.S., and used for final washing of the solid samples.

### 2.2. Hydrothermal Carbonization

A typical synthesis was carried out according to our previous work [24] as follows: 4 g of molasses was dissolved in 25 mL of deionized water and stirred for 2 h with a magnetic stirrer. Samples were then placed in Teflon-lined autoclave reactors (PARR, 4744 general acid digestion vessel—45 mL) for HTC. The digestion vessels were tightly sealed and heated in a muffle furnace at different temperatures (180–250 °C) and different times (2 h, and some samples up to 48 h). The digestion vessels were removed from the muffle furnace and allowed to cool for about 12 h. The solid products of molasses HTC were obtained after separation of the dark colored liquids by filtration and washing several times with distilled water and absolute ethanol. The recovered moist products were dried in an oven at 80 °C for 24 h. Subsequently, the obtained final products were stored in a desiccator.

### 2.3. Characterizations

Brunauer-EmmettTeller surface area (BET) was measured using a Micromeritics TriStar II PLUS (USA). Samples were degassed at 250 °C under a vacuum for 90 min while the surface areas were calculated via nitrogen adsorption/desorption isotherms [25]. Shimadzu IRPrestige-21 was used for the FTIR study to qualitatively determine the presence of certain surface functional groups. The investigation was performed in the wavenumber range of 4000–525 cm^−1^ using 16 scans at a spectral resolution wavenumber of 4 cm^−1^ [26]. The pellet samples were prepared using KBr. In order to quantify the oxygenated functional groups present on the surface of microspheres, we used a customized Boehm titration method as described in our previous report [24].

Scanning electron microscopy (SEM), Tuscan VEGA II LSU (Tuscan USA Inc.) was used to analyze the surface morphology of the products. ImageJ software, version 1.53t developed at the National Institutes of Health (NIH), USA, and freely available at http://imagej.nih.gov/ij/ (accessed on 19 September 2022), was used to analyze images from the SEM and measure the diameter of CMs at various temperatures and reaction times.

Elemental analysis (C, H, N, and S) of the samples was performed using a PerkinElmer series II CHNS/O 2400, USA analyzer. The oxygen content was taken as the remaining mass. The change in weight of the samples was measured using a TA Instrument TGA Q50 thermogravimetric analyzer. Samples were evenly and loosely distributed in an open sample pan (6.4 mm in diameter and 3.2 mm depth) with an initial sample size ranging from 6 to 10 mg. The temperature was increased from 25 to 900 °C in a controlled atmosphere [27].

The yield of the product was expressed as the final weight of the dry product. Percent yield was expressed using Equation (1):(1)$$Yield=\frac{M_{c}}{M_{o}}\times 100$$
where, *M*_c_ and *M*_o_ were the weight of the final dry product (g) and the weight of the dry raw material (g), respectively.

### 2.4. Adsorption Study

Methylene blue (MB) of 95% was used to produce a stock solution of 1000 mg/L. The pH was maintained at 6 by addition of sulfuric acid or sodium hydroxide (0.1 mol/L). The pH values of the solutions of MB were determined using a pH meter. The batch adsorption experiment was conducted in Erlenmeyer flasks with solutions of MB at different initial concentrations (25 to 200 mg/L) added to each flask. Samples (0.125 g) with a particle size less than 0.25 mm were added to the dye solution (2 mL) and placed in a rotating shaker (150 rpm) at 25 °C. For the equilibrium studies, the experiment was run for 540 min to ensure that equilibrium conditions were achieved.

During the batch adsorption process, a flask was removed from the rotary shaker several times. The supernatant solution (10 mL) was extracted and centrifuged for 5 min at 5000 rpm in a Hettich EBA 20 centrifuge (Germany) to settle suspended particles and reduce disturbance by small particles of the prepared material. Approximately 5 mL of the supernatant solution was taken from each flask and diluted with the appropriate amount of deionized water and then stored in a closed bottle for analysis using a UV-Vis spectrophotometer (Jasco V-770 UV-Visible/NIR spectrophotometer, Japan). The concentration of MB was determined by comparing the absorbance with a calibration curve. The total amount of MB adsorbed on the carbon microspheres at time *t*, $q_{t}$ (mg/g) can be determined by using the following Equation (2):(2)$$q_{t}\;=\;\frac{\left(C_{o}-C_{t}\right)\;V}{M_{c}}$$
where, $q_{t}$ (mg/g) is the equilibrium adsorption capacity, *C_o_* is the initial concentration of the solution (mg/L), *C*_t_ is the concentration of the solution at time t (mg/L), *V* is the volume of the solution (L) and *M_c_* is the weight of the dry adsorbent (g). The equilibrium adsorption capacity was determined using the following equation:(3)$$q_{e}\;=\;\frac{\left(C_{o}-C_{e}\right)\;V}{M_{c}}$$
where, $q_{e}$ (mg/g) is the equilibrium adsorption capacity and *C_e_* is the equilibrium concentration of the solution (mg/L). The removal efficiency can be estimated using the following equation:(4)$$R\left(\%\right)\;=\;\frac{\left(C_{o}-C_{e}\right)\;}{C_{o}}\times 100$$

## 3. Results and Discussions

### 3.1. Product Yield

The yield of converting the palm date molasses to products are summarized in Table 1. The obtained products were designated as “MO-X-Y”, where “X” represented the HTC temperature and “Y” for reaction time. The experiments showed that solid products were not formed when the reaction temperature and reaction time were less than 180 °C and 2 h, respectively. At these conditions, the color of the product from the hydrothermal process was brown and passed through the filter paper without any retained solid products. This result was in good agreement with the work of Li and co-researchers [28]—they did not obtain CMs from glucose at 180 °C and reaction times of less than 2 h. An increase in temperatures and reaction times leads to an increase in yield values. It is worth noting that a low temperature (e.g., 180 °C) and a low reaction time (e.g., 2 h) produce only a very small amount of CMs with a production rate of 3.6%. Increasing the reaction time from 2 to 30 h leads to an increase in CM production to 36%. Increasing the reaction time beyond 30 h does not increase the production rate, as can be seen at 36 h (Table 1). It is clear that the production rate decreased from 36 to 33%. From this table, it can be seen that the highest CM production rate (40.4%) was achieved when the temperature was raised to 250 °C and the reaction time was 6 h. Hoekman and co-researchers [29] found that for genocellulose, yield decreased from 69.1% to 50.1% when the temperature was increased from 215 °C to 255 °C. The results of this work are consistent with the findings of Hoekman and co-researchers [29]—the temperature was increased from 230 to 250 °C for 8 h. However, these results are not consistent with lower reaction times (e.g., 4 and 6 h). Figure 1 and Figure 2 show the effect of temperature and residence time on the CMs yields. It is evident from Figure 1 that for reaction time of 8 h, increasing the temperature increases the yield up to a certain point and then gradually decreases. Figure 2 reveals that the yield decreases with increasing reaction time. It should be noted that high temperatures (e.g., 250 °C) can have a negative effect on the yields of CMs. This is due to a series of dehydration and decarboxylation reactions. These reactions become more pronounced at higher temperatures, resulting in an increase in the gas phase and a decrease in solid yield [30,31,32].

### 3.2. CMs Formation

SEM images of the CMs taken at different temperatures and reaction times are presented in Figure 3. Figure 3a shows small CMs developed and dispersed on the surface for the molasses sample at 180 °C and 2 h. If the reaction time is increased, e.g., from 2 to 36 h, visible and large CMs are formed. The structure and the shape of the microspheres are very identical to those obtained from glucose by other researchers [3,6,28]. Figure 4a,b shows the effects of temperature and reaction time on the diameter of CMs. An increase in temperature promotes the formation of CMs, as can be seen in Figure 4a. These results are comparable to those reported in previous studies, which used glucose as a feedstock [3,6,28]. However, at a temperature of 250 °C and a reaction time of 8 h or more, CMs tend to fade and decay (Figure 3i). This could be due to the fact that high temperatures and long reaction times lead to the formation of secondary char [20]. ImageJ software was used to measure the diameter of CMs at various temperatures and reaction times. The material formed consisted of a few agglomerates of carbonaceous microspheres (~5–9 μm in size). This result is consistent with the findings of Sevilla and Fuertes [4], who reported CMs with a diameter of ~2–10 μm from cellulose at temperatures of 220–250 °C and a reaction time of 2–4 h. The results somewhat agree with the findings of Ryu and co-researchers [16], who obtained CMs with a diameter of 1–4 μm from monosaccharides at a temperature of 170 °C and a reaction time of 12 h. It is evident that when the reaction time was increased from 2 to 36 h at 180 °C, the diameter increased from 4.8 to 7.8 μm. Moreover, when the temperature was increased from 180 to 250 °C, the diameter of the CMs gradually increased and the shape of the spheres became more visible. The largest diameter was obtained at a temperature of 250 °C and a reaction time of 6 h. These results indicate that a temperature between 200 and 250 °C and a reaction time of 6 to 8 h are optimal to obtain an acceptable production rate of CMs with regular shapes. This observation is consistent with other glucose-derived CMs [28]. Although the results obtained in this work are somewhat consistent with those obtained by Li and co-researchers for glucose, there are some differences regarding the non-aggregation of microspheres under all conditions used in this work. Li and co-researchers [28] concluded that at temperatures of 180–230 °C and a reaction time of 7 h, most microspheres aggregated.

The BET surface areas for all samples are less than 1 m^2^/g, indicating very marginal porosity (Table 2). The FTIR spectra of the four CMs samples listed in Table 2 (with indicated textural characteristics) are shown in Figure 5. The FT-IR results show that these samples are consistent with generally produced glucose-derived hydrochar. The band at 3430 cm^−1^ is mainly due to the O–H stretching vibrations in hydroxyl or carboxyl groups [6,33]. The stretching vibrations corresponding to aliphatic C–H groups appear at 2920 cm^−1^ [4,6,34]. The bands at 1700 and 1610 cm^−1^ are attributed to the C=O and C=C, respectively, indicating the presence of aromatic rings [4,34]. The absorption band at 1510 cm^−1^ is assigned to C˗O stretching vibration in esters, ether, phenols and aliphatic alcohols [35]. The peaks below 1000 cm^−1^ bands in the 875–750 cm^−1^ are assigned to aromatic C-H out of plane bending vibrations [36,37].

Our Boehm titration results show that the total oxygenated functional groups for sample MO-250-6 amounts to 0.225 mmol/g phenolic groups, 0.205 mmol/g carboxylic groups and 0.225 mmol/g lactonic groups. As such, the presence of such groups in the microspheres would indicate the predominance of adsorption mechanisms relating to electrostatic forces and hydrogen bonding (possibly via the positively-charged amino group of methylene blue). This agrees with the findings by Duy Nguyen and co-researchers [38]. We deem that van der Waals interactions are not the governing mechanisms, since surface areas of the microspheres are comparatively low [24]. Such results tally with our FT-IR findings with observations of –OH and –COOH groups.

The results of the elemental analysis are summarized in Table 3 for the raw molasses and the four CM samples. The highest carbon content was obtained for the sample at 180 °C and 36 h. It seems clear that the longer reaction times caused an increase in the carbon content of the sample. However, for the other three samples, increasing the temperature from 200 to 250 °C caused a slight increase in the carbon content in the microspheres, which was accompanied by a decrease in hydrogen contents. The O/C and H/C atomic ratios decrease with increasing temperature, as seen in samples synthesized at 200 and 250 °C. These observations are consistent with the results reported in previous studies [38,39].

TG analysis was performed to investigate the thermal behavior of the microspheres as well as the raw molasses in which TGA profiles are shown in Figure 6. It appears that for all microsphere samples, thermal degradation trends are quite similar. For the raw molasses sample, significant weight % loss precipitously occurs at temperatures higher than 100 °C as compared to microspheres, which implies the lower thermal stability of the raw molasses. This agrees with the findings reported by Ahmed Khan and co-researchers [39], as the lower weight percentages of volatile matter and higher fixed carbon weight percentages (i.e., higher concentrations of carbon materials) are postulated factors contributing to the higher thermal stability of the microspheres. All the microspheres samples experienced a noticeable weight loss up to 150 °C, which is likely attributed to the vaporization of physically adsorbed water—i.e., thermodesorption of surface oxides. At higher temperatures, the weight loss can be attributed to the dismantling/breaking of intermolecular and aromatic molecular structures [40].

We further conducted methylene blue adsorption studies (see supporting information) by using the synthesized microspheres, which were similar to previous studies [24,41], and found that the equilibrium adsorption capacity was reached after 15 min with the maximum adsorption capacity at about 12 mg/g (removal efficiency, R (%) = 14%). Table 4 shows the comparison of our obtained adsorption capacity of 12 mg/g with the adsorption capacities of adsorbents prepared from other biomass sources.

## 4. Conclusions

Hydrothermal carbonization of inedible crystallized date palm molasses was conducted at various temperatures (180–250 °C) and reaction times (2–10 h). This study revealed that solid products did not form at 180 °C and a reaction time of less than 2 h. The highest yield of the CMs, 40.4%, was obtained at a temperature of 250 °C and a reaction time of 6 h. However, increasing the reaction time from 6 to 8 h at 250 °C resulted in a decrease in the production rate. The study also demonstrated that temperatures between 200 and 250 °C and a reaction time of 6 to 10 h were optimal to obtain an acceptable production rate of CMs with regular shapes. It was found that the diameters of the produced CMs were in the range of ~5–9 μm. The largest diameter (8.9 μm) was obtained at 250 °C and 6 h of reaction time. In addition, the product was tested for the removal of methylene blue and found to have a maximum adsorption capacity of 12 mg/g. This indicates that the material could be a good absorbent if activated by chemical or physical methods.

## Figures and Tables

**Figure 1 materials-16-01672-f001:**
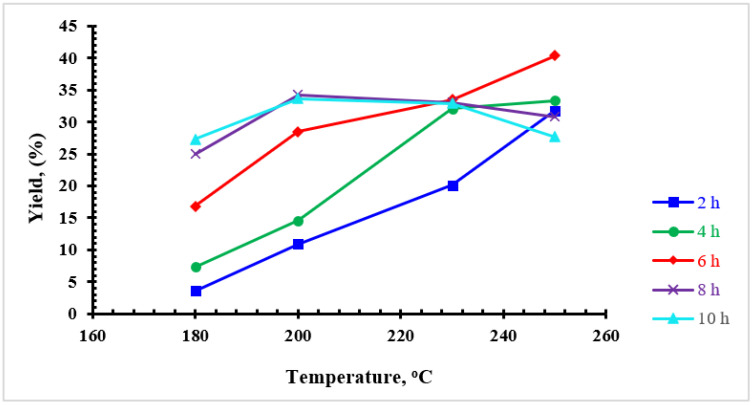
Effect of HTC temperature on the CMs yields obtained from date molasses.

**Figure 2 materials-16-01672-f002:**
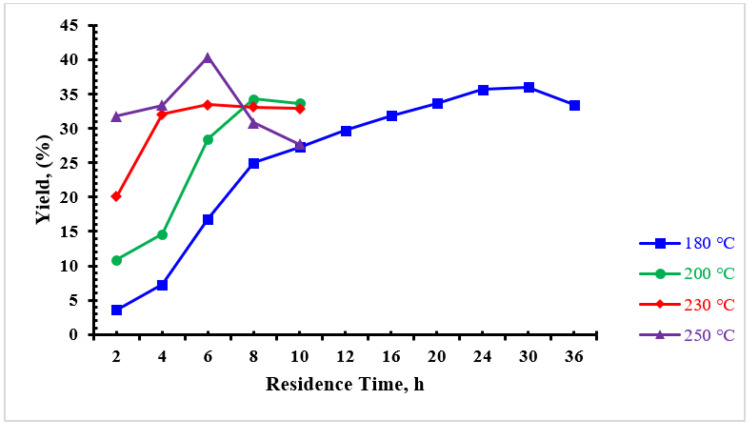
Effect of HTC reaction time on the CMs yields obtained from date molasses.

**Figure 3 materials-16-01672-f003:**
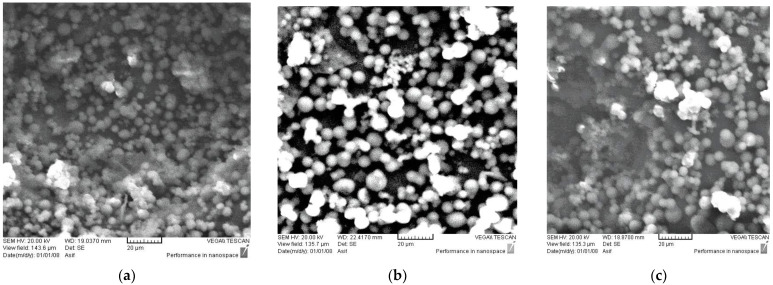
SEM images of the synthesized CMs with variations in temperature and reaction time. (**a**) Molasses at 180 °C and 2 h. (**b**) Molasses at 180 °C and 30 h. (**c**) Molasses at 200 °C and 2 h. (**d**) Molasses at 200 °C and 10 h. (**e**) Molasses at 230 °C and 2 h. (**f**) Molasses at 230 °C and 8 h. (**g**) Molasses at 250 °C and 2 h. (**h**) Molasses at 250 °C and 6 h. (**i**) Molasses at 250 °C and 8 h.

**Figure 4 materials-16-01672-f004:**
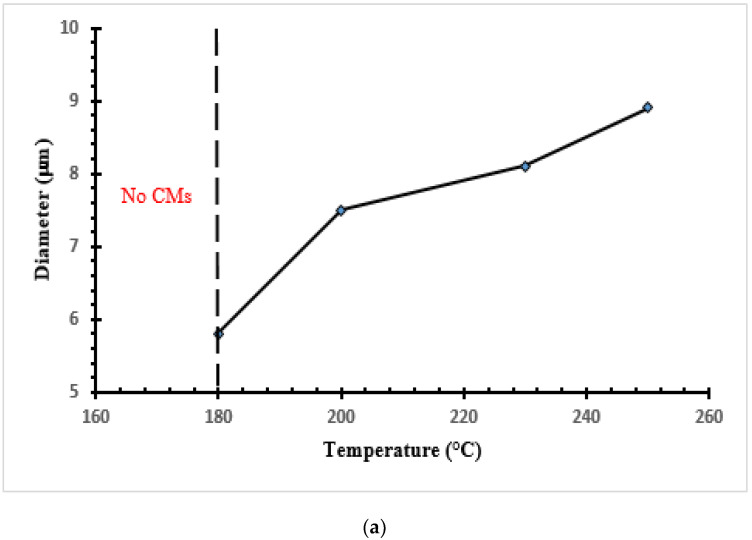
(**a**) Effects of HTC temperature on the size of CMs (reaction time = 6 h). (**b**) Effects of HTC reaction time on the size of CMs.

**Figure 5 materials-16-01672-f005:**
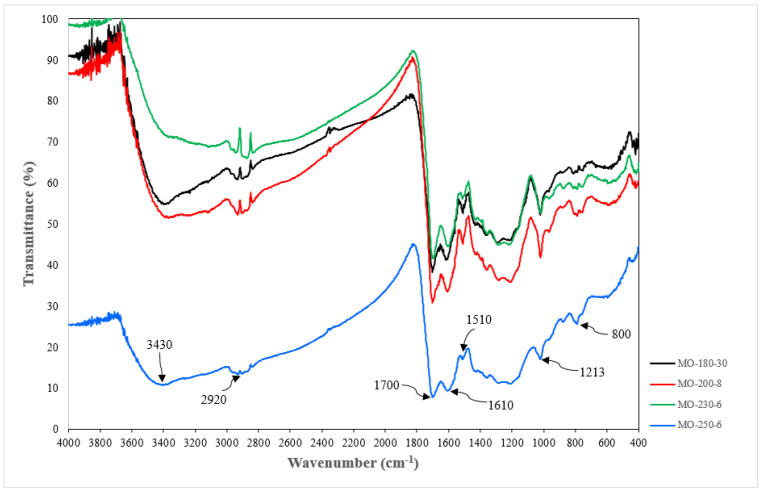
FT-IR spectra for the four CM samples.

**Figure 6 materials-16-01672-f006:**
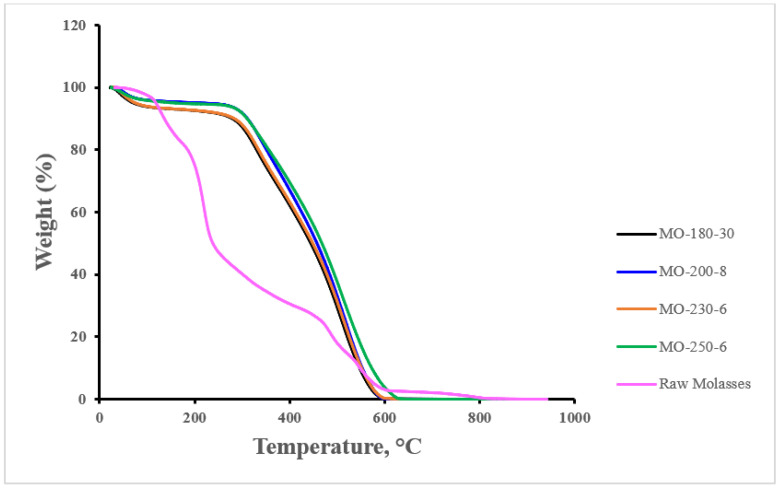
TGA profiles for raw molasses and synthesized CMs.

**Table 1 materials-16-01672-t001:** Yield of inedible crystallized date palm molasses conversion to CMs.

Sample	Yield (%)	Particle Diameter (μm)
MO-180-2	3.6	4.8
MO-180-4	7.3	5.6
MO-180-6	16.8	5.8
MO-180-8	25.0	6.3
MO-180-10	27.3	6.5
MO-180-12	29.7	6.9
MO-180-16	31.9	7.0
MO-180-20	33.7	7.2
MO-180-24	35.7	7.3
MO-180-30	36.0	7.6
MO-180-36	33.5	7.8
MO-200-2	10.9	6.5
MO-200-4	14.6	7.1
MO-200-6	28.5	7.5
MO-200-8	34.3	7.8
MO-200-10	33.7	7.9
MO-230-2	20.1	7.4
MO-230-4	32.1	7.8
MO-230-6	33.5	8.1
MO-230-8	33.1	8.2
MO-230-10	32.9	8.4
MO-250-2	31.8	7.8
MO-250-4	33.4	8.3
MO-250-6	40.4	8.9
MO-250-8	30.8	No CMs
MO-250-10	27.7	No CMs

**Table 2 materials-16-01672-t002:** Textural and pore characteristics of the synthesized carbon microsphere samples.

CM Sample	BET Surface Area (m^2^/g)	Pore Volume (cm^3^/g)	Pore Size (nm)
MO-180-30	0.7976	0.0029	48.70
MO-200-8	0.7246	0.0033	36.66
MO-230-6	0.7968	0.0023	46.31
MO-250-6	0.3500	0.0020	43.34

**Table 3 materials-16-01672-t003:** Elemental analysis results for the inedible crystallized date palm molasses and synthesized CM samples.

CM Samples	Chemical Composition					O/C (Atomic)	H/C (Atomic)
	C (wt.%)	H (wt.%)	N (wt.%)	S (wt.%)	O (wt.%)		
Molasses	34.53	7.95	0	1.08	56.44	1.23	2.76
MO-180-30	67.85	4.83	0.09	0.63	26.60	0.29	0.85
MO-200-8	64.87	4.72	0.04	0.59	29.78	0.34	0.87
MO-230-6	67.57	4.62	0.30	0.57	26.94	0.30	0.82
MO-250-6	68.22	4.60	0.18	0.56	26.44	0.29	0.81

**Table 4 materials-16-01672-t004:** Removal of methylene blue using adsorbents prepared from other biomass sources.

Biomass	BET Surface Area (m^2^/g)	Adsorption Capacity (mg/g)	Reference
Sabal palm	14.68–59.77	18–25	[24]
Sludge	31.00–11.85	4.79–37.95	[42]
Coffee husk	2.90–31.22	34.85	[43]
Inedible crystallized date molasses	0.7968–0.3500	12	This study

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
