# Peer review of "Synthesis of Carbon Microspheres from Inedible Crystallized Date Palm Molasses: Influence of Temperature and Reaction Time"

_materials, 2023, doi:10.3390/ma16041672_

Round 1
Reviewer 1 Report
The authors have shown interesting results in this manuscript: Synthesis of Carbon Microspheres from Inedible Crystallized Date Palm Molasses: Influence of Temperature and Residence time. However, there are many aspects that need to be clarified and improved. Also, there is no information in the manuscript about the supplement, explaining how the adsorption capacity of MC towards methylene blue.
It is suggested to expand/ add more data on the adsorption capacity as part of the characterization of CM also can give more information in the future use of CM.
The comments are available in the reviewed manuscript. To conclude, it is not recommended to accept this manuscript in its current state.

Author Response
We wish to express our appreciation to the reviewers for their insightful comments, which have helped us significantly to improve our manuscript. According to the suggestions, we have thoroughly revised our manuscript and its final version is enclosed. Point-by-point responses to the comments are listed below.
Reviewer #1:
Comments:
Please re-arrange the introduction. the 1st paragraph is too long.
Response:
We have rearranged the Introduction section.
Comments:
Include all materials. Dyes is not yet mentioned here, etc.
Response:
The dye (methylene blue) is added to Section 2.1 (Raw material and Chemicals).
Comments:
How the solid product can be obtained by filtration of the dark colour liquids without any precipitate or solid products?
Response:
The solid products of molasses HTC were obtained after separation of the dark coloured liquids by filtration and washing several times with distilled water and absolute ethanol.
A No. 5 sintered glass filter with No. 2 Whatman filter paper were used to ensure that no solid products were lost during the filtration process.
Comments:
Usually for research it requires more than 16 scans (128 scans).
Response:
FT-IR analysis in our study represents a qualitative assessment of the functional groups of the microspheres, hence, 16 scans are more than sufficient. The 16 scans for FT-IR are essentially a prelude to a more robust quantification of the weakly acidic functional groups as determined by the reported Boehm titration result.
Comments:
XRD measurement is required to see the structure of CM
TGA measurement is needed also to confirm the temperature of carbonization
Response:
XRD is not required because it is well known that carbon materials will provide broad and widened peaks illustrating the non-crystallinity (amorphous nature) of the microspheres as opposed to actual crystals.
We have included the CHNS and TGA results in the manuscript.
Comments:
What happened to the morphology of this CM, while the time resident was increase for 2 extra hours?
Comment on the arrangement of the image:
Please arrange it in order of the increase of resident time. in example:
2h, 6h, 8h. Fig 3a-fig 3f were not arranged in that order.
Response:
We selected only the figures for the shortest and longest residence times for each sample. For example, for samples conducted at 180 °C, the minimum time for the formation of carbon microsphere is 2 hours and the maximum time at which the yield began to decrease is 36 hours. However, at 250 °C, the highest yield was reached after 6 hours and any further increases in time led to the disappearance of the carbon microspheres.
Comments:
The surface area is too low for porous materials.
Why the SAA measurement is necessary? It is expected that CM has certain or high surface area?
In general, graphitic carbon usually porous and has high surface area.
Response:
The date molasses has more than 40% glucose (Mohamed and Babucurr, 2015). The surface area of glucose is lower than 1 m2/g (Aydıncak et al., 2012).
An intrinsic advantage of HTC is the simplicity of the method to induce hydrophilicity on the surface of the spheres via incorporation of –O, –C=O or charged functional groups into the spheres’ overall structure. Intuitively, the presence of such functional groups is construed to be favorable in terms of removal of dyes/heavy metals from wastewater. CMs have rich oxygen-containing functional groups on the internal and external surfaces. Surface functionalization can be easily accomplished via addition of a co-reactant that contains specific functional groups.
Mohamed, I.O. and Babucurr, J., 2015. Effect of date syrup on pasting, rheological, and retrogradation properties of corn starch gels. Starch‐Stärke, 67(7-8), pp.709-715.
Aydıncak, K., Yumak, T., Sınağ, A. and Esen, B., 2012. Synthesis and characterization of carbonaceous materials from saccharides (glucose and lactose) and two waste biomasses by hydrothermal carbonization. Industrial & Engineering Chemistry Research, 51(26), pp.9145-9152.
Comments:
Usually Y scale and title of FTIR spectra is located on the left of the chart.
Please set the scale of Y axis to 0-100 (%T)
Response:
We have revised the figure as instructed.
Comments:
It is recommended to expand the experiment on this adsorption of dyes using the as-prepared CM to give more information on the prospective application and it must be also supported by the explanation on its properties (characterization results).
Response:
We have included the results and corresponding discussion on the adsorption of dyes experiments in the last paragraph of section 3.2.
Comments:
Need to add more references, especially in the characterization and future application.
Make sure the references cited is update
Response:
We have cited more references as instructed by the reviewer.

Reviewer 2 Report
The paper describes the hydrothermal carbonisation of date palm molasses to form carbon microspheres. The only novelty of the paper seems to be the use of date palm molasses as a substrate. The authors analyzed the surface area and the materials' porosity, although the results' accuracy is far too high. It is known from the literature that sugar-derived carbon microspheres, in general, do not possess measurable surface areas or porosity. The authors analysed the surface chemistry of the spheres by Bohem titration, SEM and FTIR and applied the microspheres for adsorption. In general, this is a weak paper containing very little novel data which would be attractive for investigators in this field. The authors should add further characterizations of the obtained microspheres, including elemental analysis, or at least SEM-EDX to show the elemental composition of their material. Moreover, the TG studies of the functional groups and the oxidative stability would require the paper to be published in Materials. In the present form, the manuscript can be suitable for a lower-impact factor journal.
Abstract- residence time is used more for the reaction with catalysts. Here I would call it a reaction time of 6h
1) The example of the application of these carbon spheres in adsorption should be mentioned in the abstract. It is somehow surprising that the materials with such a low BET surface area can adsorb methylene blue. The adsorption results should be compared with the adsorption results of activated carbon carried out under the same reaction conditions
2) beneficial step in the application of sustainable processing industries in the Middle East.
3) Introduction-The review of different findings concerning the hydrothermal carbonization of various sugars in the introduction should be rewritten in a more organized manner. The authors mix discussion of the influence of various reaction parameters and repeat the information about the influence of the synthesis temperatures and the range of temperatures appropriate to synthesize microspheres. It is irrelevant that someone made microspheres at slightly different temperatures than other groups reported. The introduction should include the influence of the substrates, temperature, time, catalyst (if any), promotors, etc.
4) What is the novelty of this work? Is it the type of substrate?
5) Due to long or improper storage
6) The authors should check the formatting of the manuscript as there are some extra spaces in the document
7) These results are comparable to those reported in previous works which used glucose as a feedstock(Titirici et al., 2008; Li et al., 2012; Qi et al., 2015).
8) The software used for measuring the particle sizes should be listed in the experimental part of the paper
9) The results should be consistent with the ones reported for glucose-derived microspheres, because the molasses used in this work contain high amounts of glucose and fructose as the authors stated in the introduction. There is no need to comment on it.
10) Effect of HTC reaction time (in the caption for the figures)
11) How did the authors measure such a low surface area (below 1m2/g???), and what is the error of these measurements? The results given in Table 2 are far too accurate. Representative N2-isotherms (MO-250-6 and MO-180-30) should be included in the paper. After revising the literature, the authors should be aware that the hydrochars derived from sugars practically do not have any surface area, so I would say there is no porosity in these materials. How was the pore size measured if the porosity was almost zero? Are the authors sure about their measurements and their accuracy?
12) The description of FTIR should be formatted.
Author Response
We wish to express our appreciation to the reviewers for their insightful comments, which have helped us significantly to improve our manuscript. According to the suggestions, we have thoroughly revised our manuscript and its final version is enclosed. Point-by-point responses to the comments are listed below.
Reviewer #2:
Comments:
Abstract- residence time is used more for the reaction with catalysts. Here I would call it a reaction time of 6h
Response:
The term “reaction time” has been now indicated in lieu of “residence time”.
Comments:
The example of the application of these carbon spheres in adsorption should be mentioned in the abstract. It is somehow surprising that the materials with such a low BET surface area can adsorb methylene blue. The adsorption results should be compared with the adsorption results of activated carbon carried out under the same reaction conditions.
Response:
The Boehm titration results indicating 0.225 mmol/g phenolic groups, 0.205 mmol/g carboxylic groups and 0.225 mmol/g lactonic group in our microspheres would indicate the predominance of adsorption mechanisms relating to electrostatic forces and hydrogen bonding (possibly via the positively-charged amino group of methylene blue). This agrees with the findings by Nguyen and co-researchers (Nguyen et al., 2019). We deem that van der Waals interactions are not the governing mechanisms since surface areas of the microspheres are comparatively low.
Nguyen, D.H., Tran, H.N., Chao, H.-P., Lin, C.-C., 2019. Effect of nitric acid oxidation on the surface of hydrochars to sorb methylene blue: An adsorption mechanism comparison. Adsorption Science & Technology 37(7-8), 607-622.
Comments:
…beneficial step in the application of sustainable processing industries in the Middle East.
Response:
We have revised the sentence accordingly.
Comments:
What is the novelty of this work? Is it the type of substrate?
Response:
As far as we know, this is the first study in which CMs were synthesized from inedible date palm molasses. The shape and arrangement of the CMs closely resembled those obtained from glucose. This is a good indication that the material can be a good absorbent after chemical and physical treatment.
Comments:
Due to long or improper storage
Response:
We have revised the sentence accordingly.
Comments:
The authors should check the formatting of the manuscript as there are some extra spaces in the document
Response:
We have removed the extra spacing.
Comments:
These results are comparable to those reported in previous works which used glucose as a feedstock (Titirici et al., 2008; Li et al., 2012; Qi et al., 2015).
Response:
We have revised the sentence accordingly.
Comments:
The software used for measuring the particle sizes should be listed in the experimental part of the paper
Response:
We have listed the software in the experimental section.
Comments:
The results should be consistent with the ones reported for glucose-derived microspheres, because the molasses used in this work contain high amounts of glucose and fructose as the authors stated in the introduction. There is no need to comment on it.
Response:
The morphology of the microspheres is somewhat similar to the ones reported for glucose-derived microspheres.
Comments:
Effect of HTC reaction time
Response:
We have revised the caption of the figures as suggested.

Reviewer 3 Report
Review of the Manuscript No materials-2167917
1. Title.
“Synthesis of Carbon Microspheres from Inedible Crystallized Date Palm Molasses: Influence of Temperature and Residence time”
Should be replaced by
“Synthesis of Carbon Microspheres from Inedible Crystallized Date Palm Molasses: Influence of Temperature and Duration”
2. Introduction should include the main aims of the work which are planned, and the tasks that will be performed.
3. Materials and Methods.
Please add more info about IR spectroscopy measurements. Do you used KBr pellets? Does the IR spectrometer use ATR accessory?
4. Materials and Methods.
“Scanning electron microscopy (SEM), Tuscan VEGA II LSU (Tuscan USA Inc.) was used to analyse the surface morphologies of the products.”
Should be substituted with
“Scanning electron microscope (SEM) Tuscan VEGA II LSU (Tuscan USA Inc.) was used to analyse the surface morphologiy of the products.”
5. Materials and Methods
Please add info about equipment used to obtain BET surface areas of the samples.
6. I suggest the authors to perform the hydrothermal carbonization at higher temperatures, in order to obtain products with larger surfaces area.
7. Conclusions
“HTC of inedible crystallized date palm molasses was conducted at various temperatures (180 –250 °C) and residence times (2 – 10 h).”
Should be replaced by
“Hydrothermal carbonization of inedible crystallized date palm molasses was conducted at various temperatures (180 –250 °C) and duration (2 – 10 h).”
8. Authors should correct the references according to journal requirements. Please add also DOI.
Author Response
We wish to express our appreciation to the reviewers for their insightful comments, which have helped us significantly to improve our manuscript. According to the suggestions, we have thoroughly revised our manuscript and its final version is enclosed. Point-by-point responses to the comments are listed below.
Reviewer #3:
Comments:
Title:
“Synthesis of Carbon Microspheres from Inedible Crystallized Date Palm Molasses: Influence of Temperature and Residence time”
Should be replaced by
“Synthesis of Carbon Microspheres from Inedible Crystallized Date Palm Molasses: Influence of Temperature and Duration”
Response:
We have changed the title as the review requested.
Comments:
Introduction should include the main aims of the work which are planned, and the tasks that will be performed.
Response:
The last 5 lines of the introduction contain the main aim of the work.
Comments:
Materials and Methods.
Please add more info about IR spectroscopy measurements. Do you used KBr pellets? Does the IR spectrometer use ATR accessory?
Response:
FTIR spectroscopy measurements were performed using KBr pellets.
The IR spectrometer does not use ATR accessory.
Comments:
Materials and Methods.
“Scanning electron microscopy (SEM), Tuscan VEGA II LSU (Tuscan USA Inc.) was used to analyse the surface morphologies of the products.”
Should be substituted with
“Scanning electron microscope (SEM) Tuscan VEGA II LSU (Tuscan USA Inc.) was used to analyse the surface morphology of the products.”
Response:
We have made the change as requested.
Comments:
Materials and Methods
Please add info about equipment used to obtain BET surface areas of the samples.
Response:
We have added information on the equipment used to determine the BET surface areas of the samples
Comments:
I suggest the authors to perform the hydrothermal carbonization at higher temperatures, in order to obtain products with larger surfaces area.
Response:
Hydrothermal carbonization method is usually carried out at 160-250 oC. The autoclave used in this work can be used up to 250 oC.
Comments:
Conclusions
“HTC of inedible crystallized date palm molasses was conducted at various temperatures (180 –250 °C) and residence times (2 – 10 h).”
Should be replaced by
“Hydrothermal carbonization of inedible crystallized date palm molasses was conducted at various temperatures (180 –250 °C) and duration (2 – 10 h).”
Response:
We have revised the sentence accordingly.
Comments:
Authors should correct the references according to journal requirements. Please add also DOI.
Response:
The references are retyped according to the format of the journal.

Round 2
Reviewer 1 Report
Dear Authors,
The revised manuscript is much better than the original one. However, there are some minor mistakes that can be revised, as follow:
1. the last sentence in the introduction: Finally, we examined the ability of the synthesized to remove methylene blue, a known water-polluting dye.
It is unclear, so it needs to be changed to: the as-synthesized CMs
2. Sectio 2.3: .... at http://imagej.nih.gov/ij/ ; the font type is not the same with other sentences.
3. Table 1. Yield of the samples. the information is incomplete. It is needed to be changed to Yield of the Palm Date Molasses conversion to CMs
4. Fig. 3a: Molasses at 180°C and 2 h, etc.
Please just type: a: Molasses at 180°C and 2 h, b. Molasses at 180°C and 30 h below each designated SEM image. Then, below the last row of images, type: Figure 3: SEM images of as-synthesized CMs with the variation in temperature and resident time
5. Figure 5. FT-IR spectra for the four samples.
Please change the caption to Figure 5. FT-IR spectra for the four CM samples.
6. Please be consistent with the abbreviation. For example, hour is shortened to h, not hr. Also please stick to American or British English, do not mix them. For example, synthesize (American) but synthesise (British).
Should this minor but important revision been conducted, the final revised manuscript can be recommended to be accepted.
Author Response
We wish to express our appreciation to the reviewers for their insightful comments, which have helped us significantly to improve our manuscript. According to the suggestions, we have thoroughly revised our manuscript and its final version is enclosed. Point-by-point responses to the comments are listed below.
Reviewer #1:
The revised manuscript is much better than the original one. However, there are some minor mistakes that can be revised, as follow:
- the last sentence in the introduction: Finally, we examined the ability of the synthesized to remove methylene blue, a known water-polluting dye. It is unclear, so it needs to be changed to: the as-synthesized CMs.
Response:
We have revised as instructed by the reviewer.
- Sectio 2.3: .... at http://imagej.nih.gov/ij/ ; the font type is not the same with other sentences.
Response:
We have changed the font type of the link to match the other parts of the manuscript.
- Table 1. Yield of the samples. the information is incomplete. It is needed to be changed to Yield of the Palm Date Molasses conversion to CMs
Response:
We have revised the title of Table 1 as instructed by the reviewer.
- Fig. 3a: Molasses at 180°C and 2 h, etc.
Please just type: a: Molasses at 180°C and 2 h, b. Molasses at 180°C and 30 h below each designated SEM image. Then, below the last row of images, type: Figure 3: SEM images of as-synthesized CMs with the variation in temperature and resident time
Response:
We have made the change as instructed by the reviewer.
- Figure 5. FT-IR spectra for the four samples.
Please change the caption to Figure 5. FT-IR spectra for the four CM samples.
Response:
We have changed the caption of figure 5 as requested by the reviewer.
- Please be consistent with the abbreviation. For example, hour is shortened to h, not hr. Also please stick to American or British English, do not mix them. For example, synthesize (American) but synthesise (British).
Should this minor but important revision been conducted, the final revised manuscript can be recommended to be accepted.
Response:
We have made the change as instructed by the reviewer.

Reviewer 2 Report
The manuscript has been improved, however, it´s novelty is moderate to low. The only novelty of the work is the use of date palm molasses as a substrate for production of carbon materials. It can be expected that biomass materials containing sugars will be suitable as substrates to produce carbon materials via HTC (see https://www.sciencedirect.com/science/article/pii/S0048969722067274 and the references therein.
The authors should at least add some information about the adsorption capacity of these materials, e.g. the comparison of the adsorption results obtained by their materials to the ones published in the literature for carbon materials obtained from biomass. Or test other standard materials under the same reaction conditions.
Author Response
We wish to express our appreciation to the reviewers for their insightful comments, which have helped us significantly to improve our manuscript. According to the suggestions, we have thoroughly revised our manuscript and its final version is enclosed. Point-by-point responses to the comments are listed below.
Comment:
The manuscript has been improved, however, it´s novelty is moderate to low. The only novelty of the work is the use of date palm molasses as a substrate for production of carbon materials. It can be expected that biomass materials containing sugars will be suitable as substrates to produce carbon materials via HTC (see https://www.sciencedirect.com/science/article/pii/S0048969722067274 and the references therein.
Response:
The novelty can be construed as moderate. Nevertheless, we deem the research impact to be rather substantial given the fact that the agricultural/food industry focusing on data palm molasses is large in the Middle East Region – there are 31 million palm date trees in Kingdom of Saudi Arabia, alone. Therefore, the impact of our research is significant because the recycling of large amounts of waste date palm molasses to generate a useable carbon microsphere represents a beneficial step in the application of sustainable processing industries in the Middle East. Therefore, there is the advantage of economies of scale when it comes to the hydrothermal technique as applied to the large amounts of palm molasses.
Comment:
The authors should at least add some information about the adsorption capacity of these materials, e.g. the comparison of the adsorption results obtained by their materials to the ones published in the literature for carbon materials obtained from biomass. Or test other standard materials under the same reaction conditions.
Response:
The following information was added to the last paragraph of the results section.
“Our obtained adsorption capacity of 12 mg/g can be compared to adsorption capacities of 4.79 to 37.95 mg/g (hydrochar from sludge) (Ferrentino et al., 2020), 18-25 mg/g (sabal palm hydrochar) (Al-Awadi et al., 2022) and 34.85 mg/g (coffee husk hydrochar) (Ronix et al., 2017)”
